# The Potential for Targeting AVIL and Other Actin-Binding Proteins in Rhabdomyosarcoma

**DOI:** 10.3390/ijms241814196

**Published:** 2023-09-17

**Authors:** Robert Cornelison, Laine Marrah, Adelaide Fierti, Claire Piczak, Martyna Glowczyk, Anam Tajammal, Sarah Lynch, Hui Li

**Affiliations:** Department of Pathology, School of Medicine, University of Virginia, Charlottesville, VA 22908, USA

**Keywords:** rhabdomyosarcoma, pediatric cancer, actin-binding protein, advillin, AVIL

## Abstract

Rhabdomyosarcoma (RMS) is the most common pediatric soft-tissue cancer with a survival rate below 27% for high-risk children despite aggressive multi-modal therapeutic interventions. After decades of research, no targeted therapies are currently available. Therapeutically targeting actin-binding proteins, although promising, has historically been challenging. Recent advances have made this possibility more salient, including our lab’s identification of advillin (AVIL), a novel oncogenic actin-binding protein that plays a role in many cytoskeletal functions. AVIL is overexpressed in many RMS cell lines, patient-derived xenograft models, and a cohort of 30 clinical samples of both the alveolar (ARMS) and embryonal (ERMS) subtypes. Overexpression of AVIL in mesenchymal stem cells induces neoplastic transformation both in vitro and in vivo, and reversing overexpression through genetic modulation reverses the transformation. This suggests a critical role of AVIL in RMS tumorigenesis and maintenance. As an actin-binding protein, AVIL would not traditionally be considered a druggable target. This perspective will address the feasibility of targeting differentially expressed actin-binding proteins such as AVIL therapeutically, and how critical cell infrastructure can be damaged in a cancer-specific manner.

## 1. Introduction

Rhabdomyosarcoma (RMS) is the most common cancer of soft tissue in children, accounting for at least half of all pediatric sarcomas [1]. Approximately 90% of all RMS cases occur in patients under 25 years of age, with the majority occurring in children under 10 [2]. RMS has significant variability in outcomes. Low-risk patients with localized disease face a survival rate of 90%, while high risk patients with metastatic or recurrent disease face dismal survival rates of 21% and 30%, respectively [3,4]. These high-risk groups have seen no improvement in outcomes in the last 30 years, making the lack of novel targeted treatments a major roadblock to durable cures.

The major anatomic sites of RMS include the head, neck, and genitourinary tract. Previously, the classification of RMS was primarily based upon histology to delineate alveolar RMS (ARMS) and embryonal RMS (ERMS). The most recent World Health Organization (WHO) classification of RMS consists of four groups: alveolar, embryonal, pleiomorphic, and spindle cell/sclerosing, with new subdivisions of spindle cell/sclerosing tumors based upon molecular alterations (Table 1) [5,6,7]. These alterations differentially regulate the pathways involved in promoting tumorigenesis. For instance, PAX3/7 and FOXO1 function as transcriptional activators of genes that regulate muscle differentiation, cell lineage, proliferation, cell migration, muscle growth, and metabolism [8,9]. Chromosomal translocations can lead to *PAX3-FOXO1* or *PAX7-FOXO1* gene fusions, and the translated fusion proteins have enhanced transcriptional activity in contrast to the parent proteins [10]. In ERMS, the *RAS* genes encode small GTPase transductor domains involved in regulating cell proliferation, migration, and growth by cycling between GTP-bound active and GDP-bound inactive forms. Mutations in these genes lead to a gain-of-function in the protein with a constitutive GTP-bound active form, promoting carcinogenesis [11]. Additionally, p53 is a tumor suppressor protein that functions to negatively regulate cell cycle progression and is frequently mutated in ERMS [12]. Most ERMS present with loss-of-function mutations in p53, which allows for uncontrolled cell cycle progression [12]. Several other functional genetic aberrations that promote RMS were recently reviewed [13]. Table 1 provides an overview of the different genetic profiles, histopathologies, and common sites associated with the four groups of RMS based on the WHO classifications.

Frontline therapy is the same for all RMS risk groups and is generally a combination of surgery, radiotherapy, and a three-drug cytotoxic chemotherapy regimen consisting of vincristine sulfate, dactinomycin (Actinomycin-D), and cyclophosphamide (VAC) in the USA, with ifosfamide replacing cyclophosphamide (IVA or VAI) in Europe [14,15]. The clinical trial results for traditional chemotherapy regiments are mixed, with some showing marginal increases in overall survival (OS) and disease/event free survival (DFS/EVS), and some showing no statistically significant difference between the two groups (Table 2). Despite some positive outcomes, traditional chemotherapy creates numerous deleterious side effects due to its non-specific cytotoxicity. For low-risk patients with localized disease, attempts to decrease the dosing of radiotherapy and chemotherapy, specifically cyclophosphamide, have been made to reduce deleterious toxicities such as lifetime infertility and myelosuppression [16,17]. With no targeted therapies approved, high-risk children face lifelong consequences even if a durable cure is achieved.

Efforts have been made to pharmacologically inhibit molecular targets in RMS, such as platelet-derived growth factor receptors (PDGFRs) and vascular endothelial growth factor receptors (VEGFRs), but phase II clinical trials have indicated no improved outcomes (Table 2). The lack of approved targeted therapies is a major unmet need in the field, and novel therapeutic targets may be necessary to decrease cytotoxic chemotherapy dosing and reduce long-lasting toxicities. Our lab has recently discovered AVIL, an actin-binding protein, as an oncogenic driver of RMS. This review summarizes the current literature on research into targeting actin-binding proteins including AVIL and its family members, and explores the therapeutic potential of drugging actin-binding proteins such as AVIL for precision oncology applications in RMS.

## 2. Improving VAC Chemotherapy and Molecular Targeted Therapies

The standard management of RMS includes chemotherapy, radiation therapy, and tumor resection. In patients with metastatic RMS, efforts toward complete remission with high-dose chemotherapy resulted in treatment-related adverse effects. Several phase III clinical trials have been conducted to improve disease-free survival, including the addition of low-dose maintenance chemotherapy after standard chemotherapy with IVA, which showed a modest increase in the 5-year disease-free survival rate of 8% [18] (Table 2). The addition of doxorubicin, a drug widely used to treat soft-tissue sarcoma, to IVA chemotherapy failed to improve the 3-year event-free survival and also resulted in treatment-related adverse effects such as infections, anemia, leukopenia, gastrointestinal disorder, and even death [19]. Another phase III clinical trial attempted to modify a similar standard therapy, VAC, to reduce toxicity by substituting half of the VAC course with vincristine and irinotecan; this resulted in no significant difference in the oncological outcome [20]. Efforts have been made to increase complete remission and disease-free survival; however, no recent trials have significantly prolonged patient survival and remission. High dose chemotherapy, low-dose maintenance chemo following standard chemo, and the addition of doxorubicin and irinotecan were all evaluated in clinical trials and found to be ineffective.

Molecular targeted therapies have also been investigated, but their efficacy is still in question. A phase II clinical trial demonstrated that patients receiving vinorelvine (V), cyclophosphamide (C), and bevacizumab experienced a response rate of 32%, while patients receiving VC and temsirolumus experienced an increased response rate of 47% [24]. Despite promising preclinical evidence in their efficacy against RMS, sorafenib and crizotinib both proved inactive against RMS in phase II clinical trials [21,22,25]. Vismodegib trials in advanced chrondrosarcoma showed some efficacy, a 25.6% clinical benefit in patients, which was short of the 40% clinical benefit goal [26,27]. Overall, there is immense genetic, biological, and clinical response heterogeneity common to RMS, underscoring the dire need for some form of targeted therapy to open new therapeutic avenues and reduce the toxicities seen in the current standard of care.

## 3. Targeting Actin and Actin-Binding Proteins

Eukaryotic actin is a 375 amino acid polypeptide that folds into four subdomains, with an ATP-binding cleft important for regulating the dynamic switch between its two forms: globular actin (G-actin) and filamentous actin (F-actin). These processes are critical to the physiological functions of cells including cell division, maintaining structural integrity, cell migration, vesicular trafficking, cell signaling, cell adhesions, and tight junction formation. While healthy cells depend on the controlled regulation of actin to maintain cellular function, cancer cells harness these same mechanisms to promote migration, invasion, and metastasis by dysregulating actin-binding proteins (ABPs) and disrupting the balance in actin dynamics, thereby facilitating the formation of invasive structures like lamellipodia and filopodia [28]. As such, efforts to therapeutically target ABPs in cancer have been an attractive but challenging field of research over recent decades.

Although targeting ABPs confers an advantage to targeting cancer cells, normal cells become vulnerable to high and unbearable toxicity. For example, while cytochalasins have exhibited promising effects on breast, lung, and prostate cancers, congestion necrosis in rats has been reported at the edge of the liver, as well as negative affects on cardiac contractility [29,30,31,32]. Also, chaetoglobosin has been shown to be lethal at a dose of 2 mg/kg in rats and induce spermatocyte degeneration in mice [33,34]. Furthermore, Jasplakinolide, latrunculin, and MKT-077 are other inhibitors targeting the actin cytoskeleton with anti-cancer effects and have been shown to induce cardiac toxicity, chronic seizures in rats, and retinal toxicity in humans [35,36,37,38]. ABP targeting as a treatment option has struggled as a result of this widespread toxicity.

## 4. Targeting Nucleation Factors

ABPs that nucleate and mediate branching include the Arp2/3 complex with nuclear-promoting factors, the formin family of proteins, and the tandem monomer binding nucleators. Arp2/3 has been shown to be overexpressed in several cancers including gastric, glioma, breast, lung, and colorectal cancer, where it facilitates cancer pathogenesis, growth, and invasion [39,40,41,42,43].

A major activator of Arp2/3 is the Wiskott–Aldrich Syndrome protein (WASp) nuclear-promoting factor expressed in hematopoietic stem cells. WASp has been targeted with small molecule compound #13 (SMC #13), which has a bioavailability score of 0.5, a molecular weight of 461.6 g/mol, four hydrogen bond acceptors, high gastrointestinal absorption, and drug-likeness potential [44]. Investigators have shown that SMC #13 directly interacts with WASp to promote its degradation via ubiquitination, which significantly attenuates WASp-dependent actin dynamics in SMC #13 treated hematopoietic malignancies with fewer toxicities in healthy naïve cells. Nolen and colleagues have also drugged Arp2/3 using small molecule inhibitors CK-0944636 and CK-0993548. Their experiments suggest that CK-0944636 binds between Arp2 and Arp3, potentially inhibiting the movement of the complex into their active conformation, while CK-0993548 modifies the confirmation of Arp3 by binding in the hydrophobic domain [45], thereby inhibiting actin polymerization.

Formins have also been of interest as a potential target in cancer. These proteins contain the highly conserved formin homology 2 (*FH2*) domain for facilitating actin assembly. Formins have been shown to be overexpressed in colorectal cancer [46], induce an epithelial-mesenchymal transition to facilitate colorectal carcinoma invasion [47], and regulate cell migration and metastasis in colorectal carcinoma [48]. Silencing the formin like 2 (*FMNL2*) gene has been shown to slow the growth of gastric cancer cells, demonstrating its therapeutic potential [49]. A small molecule inhibitor of the *FH2* domain (SMIFH2) was the first to be discovered to drug formins over a decade ago, both in vitro and in vivo. SMIFH2 disrupts formin-dependent actin dynamics and has no effect on Arp2/3-dependent actin dynamics [50].

## 5. Targeting Actin Polymerization and Depolymerization

Several drugs have been described to stabilize F-actin and thereby inhibit depolymerization. While these drugs may have some anti-tumorigenic activities [51], their inability to selectively target cancerous cells limits their usefulness to research purposes only. Jasplakinolide is a membrane-permeable cyclo-depsipeptide isolated from the marine sponge *Jaspis* sp. [52,53]. It binds to F-actin to stabilize polymerization, which impairs cell migration and the protrusion of lamellipodia [51]. Jasplakinolide competes with phalloidin, another depolymerization-inhibiting compound, for F-actin binding and stabilization [54]. Phalloidin, however, is membrane impermeable.

Other F-actin stabilizing drugs include doliculide [55], chondramides, and dollastin 11 [56]. While these drugs enhance actin polymerization, other drugs inhibit polymerization. Cytochalasins are membrane-permeable fungal metabolites that bind to the barbed ends of F-actin to inhibit actin polymerization [57]. Cytochalasin D has been shown to induce the hydrolysis of ATP in G-actin dimers to inhibit F-actin assembly, and, eventually, cell migration and proliferation. While the potential benefits of cytochalasins as a supplement to improve chemotherapies have been exploited [58] and reviewed [59], their safety in patients remains to be seen.

Furthermore, there are several other small molecule inhibitors targeting other actin-binding proteins that have been extensively reviewed [60]. These have shown promising effects in some actin-binding targets traditionally considered “undruggable”, such as Rho-GTPases. The ABP AVIL is of particular interest among actin-binding proteins because it is significantly overexpressed in RMS cancer cells while showing low level expression in very few normal cells [61]. Amplification at the AVIL locus has been shown not only in RMS but in other sarcomas as well, suggesting broader oncogenic properties. This selective overexpression of AVIL in RMS makes cancer cells vulnerable to targeting of the cytoskeleton while sparing actively dividing but healthy cells.

## 6. Advillin Background

Advillin (AVIL), encoded by the *AVIL* gene, is a calcium-regulated actin-binding protein originally identified in the adult murine brain [62,63]. Advillin is a member of the gelsolin protein superfamily, sharing 65 to 75% homology with adseverin [62]. Proteins of this superfamily, which also includes villin, gelsolin, and adseverin, regulate actin organization [62,63,64]. Advillin contains six homologous domains termed gelsolin-like (G1–G6) which are conserved within gelsolin superfamily members (Figure 1) [65]. The G1 and G2 domains allow for binding to phosphatidylinositol 4,5-bisphosphate (PIP2) and regulation of monomeric actin (G-actin), while G1, G2, and G3 domains can also function in severing and capping actin filaments. The G1 and G4 domains bind actin monomers, while the G2 domain is responsible for filamentous actin (F-actin) and tropomyosin binding [64,65]. The carboxy-terminus headpiece domain present in advillin, villin, villin-like protein, supervillin, and flightless 1 enables actin filament bundling. In the absence of calcium ions, the G6 and HP domains inhibit the function of G1–G3 domains [64,65].

Although the precise function and impact on signal transduction remains unclear, advillin seems to be involved in many processes, playing a vital role in neurite outgrowth and the development of neuronal cells that form ganglia [62,66]. Moreover, in normal physiology, advillin is rarely expressed outside of sensory neurons during development, or non-peptidergic nociceptors in dorsal root ganglia, the Merkel cells of the skin, and tuft cells in the gastrointestinal and biliary tracts in adulthood [67,68]. Expression in the dorsal root ganglia is restricted to isolectin B4-positive neurons, and may be involved in growth cone formation, axonal regeneration, and neuropathic pain [69].

Due to its key role in the organization of the actin cytoskeleton, which affects polarity, movement, cell division, and trafficking, it is not surprising that advillin appears to play a role in many cancers [70]. Figure 2 summarizes the general interactions between AVIL and actin, demonstrating that AVIL can modulate many actin-based processes. The *AVIL* gene is overexpressed in nearly 100% of glioblastomas and was identified as a bona fide oncogene that is crucial for glioblastoma tumorigenesis [71]. In recent studies, we have shown that AVIL expression is abnormally upregulated in RMS, where silencing this gene results in a dramatic reduction in proliferation and migration, killing cancer cells and preventing tumor formation [61]. Therefore, AVIL seems to be a viable therapeutic target in glioblastoma and RMS.

## 7. AVIL Functionality in ERMS and ARMS Subtypes

Embryonal RMS (ERMS) affects mostly children, with the most common sites of presentation being the head, neck, and genitourinary tract [9]. ERMS presents varying degrees of differentiation, from well-differentiated neoplasms to poorly differentiated tumors [72]. Anaplastic cells in some cases of RMS have been characterized by a significant hyperchromasia. Several genetic alterations have been suggested to drive the pathogenesis of ERMS, including enhanced RAS signaling with mutations in *KRAS*, *NRAS*, and *HRAS*; activation of Hedgehog (Hh) signaling; and inactivation of p53 and Rb pathways [73]. Alteration in each of these pathways allows the cells to evade growth suppressors and apoptosis, promoting proliferation. Copy number alterations have also been observed, including gain of chromosome 8 and loss of chromosomes 10 and 15 [72].

Alveolar RMS (ARMS) affects mostly adolescents and young adults. While ARMS can arise from any part of the body, it is commonly observed in perineal and paraspinal regions as well as the extremities. Pathological features include poorly differentiated rhabdomyoblasts with enlarged nuclei and scant cytoplasm [2]. Tumor cells are usually nested in fibrovascular septa and have loosened intercellular connections, creating alveolar or slit-like spaces. ARMS is characterized by the expression of diffused MYOD1 and myogenin. About 80–90% of ARMS are associated with recurrent Forkhead Box O1 (*FOXO1*) fusions. *FOXO1* forms a fusion with *PAX3* or *PAX7*, resulting in altered expression, localization, and function compared to wild-type *FOXO1* [74]. Functionally, the *PAX3–FOXO1* fusion joins the DNA-binding region of *PAX3* with the transactivation domain of *FOXO1*, creating a novel transcription factor that can activate cellular pathways that promote oncogeneic hallmarks such as rampant proliferation and evasion of apoptosis [74]. The PAX7–FOXO1 fusion product exhibits similar functionality in ARMS [75].

AVIL is thought to activate the RAS signaling pathway, which is a major cell proliferation pathway and a hallmark of ERMS. Interestingly, an oncogenic cooperativity assay found no difference in foci formation between overexpressing AVIL alone, RAS alone, or both [61]. Furthermore, AVIL overexpression in mesenchymal stem cells is sufficient to differentially express many RAS targets, increase p-MEK1/2 and p-ERK1/2, and mimic published gene signatures for RAS signaling. AVIL overexpression also appears to mimic PAX3–FOXO1 fusion signaling in ARMS, with AVIL leading to the differential expression of PAX3–FOXO1 fusion targets. AVIL is reported to be significantly expressed in many cell lines of both ARMS and ERMS, mimicking pathways upstream of both RMS subtypes including PAX3–FOXO1 and RAS. It may, therefore, serve as a connecting node for the major pathways associated with ARMS and ERMS [61]. Given its differential overexpression in numerous cancers including RMS, and its ability to activate both RAS and PAX3–FOXO1 signaling pathways, AVIL demonstrates promising therapeutic potential as a target for both ERMS and ARMS.

## 8. Perspective

Cytotoxic chemotherapies targeting the cytoskeleton are some of the most potent therapeutics for most cancers, with almost all approved therapies targeting tubulin [76,77,78]. These cytotoxic agents also come with a large list of deleterious toxicities given that they target all dividing cells, both normal and malignant. Identifying a cancer-specific cytoskeletal protein that is differentially targetable, especially in rarer malignancies, would be a major step forward in treating these diseases. AVIL appears to be a relatively cancer-specific cytoskeletal protein overexpressed in RMS, a pediatric cancer with limited treatment options. Targeting AVIL sits at the crossroads of two therapeutic philosophies: broad range cytotoxic agents that target critical cellular infrastructure common to all dividing cells, and a targeted cancer-specific therapy that limits toxicities in normal cell populations. It is this unique combination that positions AVIL as an attractive druggable target for RMS therapies.

## Figures and Tables

**Figure 1 ijms-24-14196-f001:**
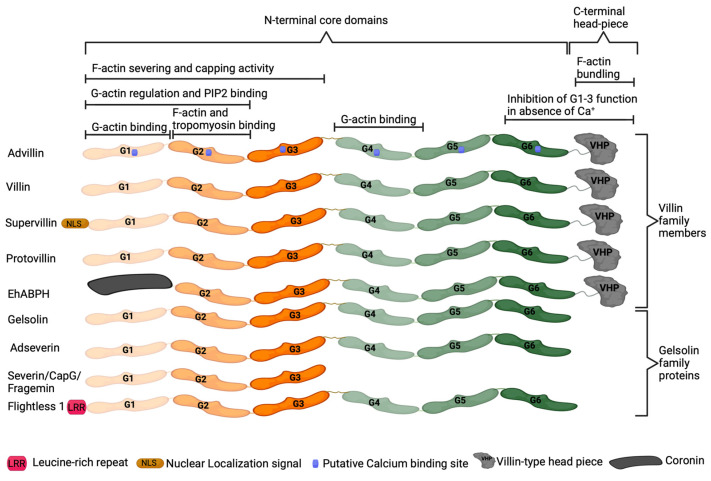
Structure of gelsolin superfamily proteins. The gelsolin-like (G1–G6) domains are largely conserved within gelsolin superfamily proteins, with domain function conserved between proteins.

**Figure 2 ijms-24-14196-f002:**
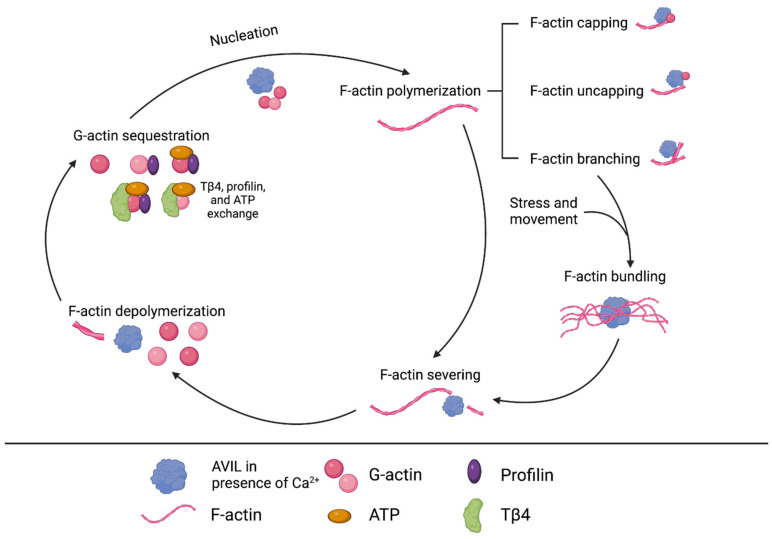
Overview of known interactions between AVIL and actin, including nucleation, capping, uncapping, branching, bundling, severing, and depolymerization. Actin is a key protein of the cytoskeleton, and its dynamic qualities enable cells to respond to change. It exists in globular (G) and filamentous (F) forms and switches between the two via polymerization or depolymerization, affecting cytoskeletal structure and cellular processes. AVIL can cause changes in actin structure and composition, affecting cellular functions like movement and division.

**Table 1 ijms-24-14196-t001:** Overview of the classification of rhabdomyosarcoma subtypes. The genetic alterations and histopathological characteristics of each of the four types are described.

Subtypes of RMS	Gene Alterations/Fusions	Histopathology	Common Sites
**Alveolar RMS** **Fusion-positive**	*MARS-AVIL* (12q14) *PAX3-FOXO1* t(2;13) *PAX7-FOXO1* t(1;13) *PAX3-NCOA1* t(2;2) *PAX3-NCOA2* t(2;8) *PAX3-IN080D* t(2;2) *PAX3- FKHR* t(2;13) *PAX7-FKHR* t(1;13)	Enlarged nuclei with scanty cytoplasm of rhabdomyoblast, not well-differentiated. Characterized by the expression of diffused MYOD1 and myogenin; 80–90% associated with recurrent *FOXO1* fusions.	Perineal region Paraspinal region Extremities
**Embryonal RMS** **Fusion-negative**	Mutations in *KRAS*, *NRAS*, and *HRAS* Aneuploidy Activation of Hedgehog (Hh) signaling Inactivation of the master regulator of p53 and Rb pathways *FGFR4* mutation *PIK3KA* mutation *NF1* mutation *FBXW7* mutation	Varying degrees of skeletal muscle differentiation with moderate cellularity.	Head Neck Genitourinary tract
**Sclerosing/Spindle Cell RMS**	*VGLL2/NCOA2* gene fusions *MYOD1* gene mutation	Fascicles of spindle cells. Elongated and fusiform nuclei, small nucleoli. Eosinophilic cytoplasm.	Testicular area Head Neck Trunk (*MYOD1* mutation)
**Pleomorphic RMS**	Complex alterations	Pleomorphic rhabdomyoblasts.	Extremities

**Table 2 ijms-24-14196-t002:** Overview of several recent clinical trials for traditional chemotherapy and targeted molecular therapy treatments for RMS.

Type of Therapy	Regimen	Phase	Patient Group	Disease/Event-Free Survival (DFS/EVF)	Overall Survival (OS)	References
**Chemotherapy**	IVA + maintenance chemotherapy	III	371 patients with non-metastatic RMS	5-year DFS With maintenance chemotherapy: 77.6% (95% CI 70.6–83.2) Without maintenance chemotherapy: 69.8% (95% CI 62.2–76.2)	5-year OS With maintenance chemotherapy: 86.5% (95% CI 80.2–90.9) Without maintenance chemotherapy: 73.7% (65.8–80.1)	[18]
**Chemotherapy**	IVA + Doxorubicin	III	484 patients with non-metastatic RMS	3-year EFS With Doxorubicin: 67.5% (95% CI 61.2–73.1) Without Doxorubicin: 63.3% (56.8–69.0) *p*-value: 0.33		[19]
**Chemotherapy**	VAC or VAC/VI	III	488 patients with intermediate-risk RMS	4-year EFS VAC: 63% VAC/VI: 59% *p*-value: 0.51	4-year OS VAC: 73% VAC/VI: 72% *p*-value: 0.80	[20]
**Molecular targeted drugs**	Sorafenib, Inhibitors of PDGFRs, VEGFRs, and MAPK	II	20 participants presenting both RMS and Wilms tumor	No objective response		[21]
**Molecular targeted drugs**	Crizotinib, inhibitors of MET, ALK, ROS1, and RON	II	13 patients with advanced and metastatic ARMS	No clinically relevant efficacy as a single agent		[22]
**Molecular targeted drugs**	Temsirolimus	II	16, 17, and 19 patients with RMS, high-grade glioma, and neuroblastoma, respectively	No clinically meaningful efficacy as a single agent in RMS		[23]

IVA: Ifosfamide, vincristine, and dactinomycin; VAC: vincristine sulfate, dactinomycin, and cyclophosphamide; VI: vincristine and irinotecan; VAC/VI: VAC course substituted by half IV; PDGFRs: platelet-derived growth factor receptors; VEGFRs: vascular endothelial growth factor receptors; MAPK: mitogen-activated protein kinase; ALK: anaplastic lymphoma kinase; RON: recepteur d’Origine Nantais; ROS1: ROS proto-oncogene 1 receptor tyrosine kinase.

## Data Availability

No new data were created or analyzed in this study. Data sharing is not applicable to this article.

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
