# Peer review of "The Potential for Targeting AVIL and Other Actin-Binding Proteins in Rhabdomyosarcoma"

_ijms, 2023, doi:10.3390/ijms241814196_

Round 1

Reviewer 1 Report

In the manuscript, the authors systematically reviewed the recent progress regarding research on advillin (AVIL) and the treatment of Rhabdomyosarcoma (RMS). Particularly, the authors discussed the potential of targeting differentially expressed actin-related proteins, including AVIL, for therapeutic purposes. Overall, the manuscript is informative, well-written, and it presents attractive druggable targets that worth further exploration to treat cancer including RMS. However, the authors seem to get lost in the manuscript discussing about AVIL and RMS, and abruptly shifted to actin-related proteins for targeted therapeutic explorations. The first four subtitles focused on the RMS and AVIL, while the remaining part of the manuscript shifted to actin-related proteins and summarized the therapeutic potential of targeting actin-related proteins for cancer treatment. The logic seems to be not clear enough to constantly catch the reader’s attention. Meanwhile, the authors intended to emphasize the therapeutic potential of targeting AVIL for treatment of RMS. However, due to limited research available about AVIL, the authors had to ‘borrow’ support from research about actin-related proteins for cancer treatment. For this reason, the authors failed to deliver the purpose. To improve the manuscript, the authors should increase discussion about AVIL and its functional involvement in tumorigenesis/RMS if such information is available, and reorganize the manuscript and modify the wording to make the logic clearer and smoother. Or the authors can downplay the discussion of AVIL, and primarily focus on actin-related proteins and their druggable potential in treatment of RMS.

In addition, the following minor issues should be addressed before considering for publication.

1.      In line 64, the authors should provide full expressions for PDGFRs and VEGFRs, although such full expressions are provided in the Table 2 description.

2.      In the Table 2 description (lines 72-76), it is recommended to move the abbreviations/full expressions to the bottom of the table for clarity. Meanwhile, the authors should organize the table to make it more readable (for example, increase space after each classification, add lines to separate different classifications, etc.).

3.      For Figure 2, the authors should provide more description in the manuscript to make the readers easier to understand the diagram.

Reviewer 2 Report

This is a well-written perspective. In this essay, the author reviewed the current status of RMS therapy and illustrated the rational of AVIL as a novel RMS therapeuctic target. Moreover, two optential treatment on AVIL were disccuessed, which should provide very useful insights on AVIL targetted RMS therapy. Overall, this perspective presented a novel druggable target for RMS, and it should bring new insights and options for cancer treatment. Overall, this perspetive should be accepted in current form.

Reviewer 3 Report

This paper reviews what is known about advillin and explores whether it can be a druggable target for rhabdomyosarcoma.  The paper seems to include the right amount of background information on rhabdomyosarcoma and its genetic alterations and treatment regimens, as well as information on advillin structure, function, and potential role in carcinogenesis.  I found the paper to be well organized and well written, as well as convincing in its attempt to highlight the potential for advillin to be a druggable target for cancer that should be explored further.  I have only two minor revision requests that would help to make the paper more clear.

(1) Line 107, there is a reference to Fig. 3, but there is no Figure 3 in the paper. I assume this is meant to be referring to Figure 3 in reference 32, since it is presented as "...(Fig. 3) [32]."  But it is a little confusing to the reader, who may search the paper for a Figure 3. Perhaps this can be presented as "...(see Fig. 3 in [32])." to make it clear where this Fig. 3 is located.

(2) Line 148, the paper states that AVIL is expressed in many ARMS and ERMS cell lines, "mimicking pathways upstream of both RMS subtypes..." I assume this means upstream pathways such as RAS? But it is a little unclear as to what is meant here by upstream pathways, so can this be clarified to list some examples, such as RAS, etc., just to make it very clear to the reader what is meant here? 

Reviewer 4 Report

The perspective by Cornelison et al. addresses the potential of targeting acting binding proteins in rhabdomyosarcoma. Specifically, they discuss the oncogenic properties of advillin in this disease. While the role of advillin in rhabdomysarcoma was clearly demonstrated in their previous PNAS paper, targeting of advillin with specific drugs seems difficult due to the expected toxicities of  drugs based on advillin expression in normal tissue. 

However, it seems worth while to further discuss targeting approaches within the scientific comunity with the help of this perspective.

Round 2

Reviewer 1 Report

In the revised manuscript, the authors made many positive adjustments. As a result, the manuscript has been greatly improved in its logic and overall quality. It is now much easier and smoother to read the manuscript.

There is one more suggestion for the authors before its publication—the authors are encouraged to add more description for Figure 2. In the current manuscript, there is only one sentence introducing Figure 2. I believe there are many readers who are not familiar with the dynamic actin polymerization and depolymerization process and the role of AVIL in the process. Inclusion of the description about Figure 2 will obviously strengthen the manuscript and benefit the readers.

Author Response

Thank you for your thoughts, we agree that we should add more information for those who are not as familiar with actin. We have added to the description of Figure 2 to include more information about actin-based processes.

Thanks,

Laine Marrah

on behalf of the authors of The Potential for Targeting AVIL and Other Actin-Binding Proteins in Rhabdomyosarcoma